# A Putative long-range RNA-RNA interaction between *ORF8* and *Spike* of SARS-CoV-2

Okiemute Beatrice Omoru[1], Filipe Pereira[2,3], Sarath Chandra Janga[1,4,5], Amirhossein Manzourolajdad[1,6]*

**1** Department of Biohealth Informatics, School of Informatics and Computing, Indiana University Purdue University, Indianapolis, IN, United States of America, **2** Centre for Functional Ecology, Department of Life Sciences, University of Coimbra, Coimbra, Portugal, **3** IDENTIFICA Genetic Testing, Maia, Portugal, **4** Department of Medical and Molecular Genetics, Indiana University School of Medicine, Medical Research and Library Building, Indianapolis, Indiana, United States of America, **5** Centre for Computational Biology and Bioinformatics, Indiana University School of Medicine, 5021 Health Information and Translational Sciences (HITS), Indianapolis, Indiana, United States of America, **6** Department of Computer Science, Colgate University, Hamilton, NY, United States of America

* amanzour@colgate.edu

**Data Availability Statement:** Yes, all data fully available without restriction. S1 Table provides GISAID accession numbers for sequences used in this study. The sequences are publicly available in the GISAID database for download.

## Abstract

SARS-CoV-2 has affected people worldwide as the causative agent of COVID-19. The virus is related to the highly lethal SARS-CoV-1 responsible for the 2002–2003 SARS outbreak in Asia. Research is ongoing to understand why both viruses have different spreading capacities and mortality rates. Like other beta coronaviruses, RNA-RNA interactions occur between different parts of the viral genomic RNA, resulting in discontinuous transcription and production of various sub-genomic RNAs. These sub-genomic RNAs are then translated into other viral proteins. In this work, we performed a comparative analysis for novel long-range RNA-RNA interactions that may involve the *Spike* region. Comparing *in-silico* fragment-based predictions between reference sequences of SARS-CoV-1 and SARS-CoV-2 revealed several predictions amongst which a thermodynamically stable long-range RNA-RNA interaction between (23660–23703 *Spike*) and (28025–28060 *ORF8*) unique to SARS-CoV-2 was observed. The patterns of sequence variation using data gathered worldwide further supported the predicted stability of the sub-interacting region (23679–23690 *Spike*) and (28031–28042 *ORF8*). Such RNA-RNA interactions can potentially impact viral life cycle including sub-genomic RNA production rates.

## Introduction

Severe acute respiratory syndrome coronavirus 2 (SARS-CoV-2) is a highly transmissible and pathogenic coronavirus that emerged in late 2019 and has caused a pandemic of acute respiratory disease, named coronavirus disease 2019 (COVID-19) [1]. SARS-CoV-2 is related to SARS-CoV-1, a life-threatening virus responsible for an outbreak in 2002–2003 that was contained after intense public health mitigation measures [2]. The Coronaviruses belong to the *Coronaviridae* family. They are enveloped, positive-sensed, and have a single-stranded RNA genome [3] and are categorized into different genera based on their protein sequences [4].

**Funding:** This research was funded by the National Institute of General Medical Sciences of the NIH under Award Number R01GM123314 (SCJ), Indo-U.S. Science and Technology Forum under Award Number IUSSTF/VN-COVID/005/2020 (SCJ) and IUPUI's Office of the Vice Chancellor for Research COVID-19 Rapid Response Grant (SCJ). There are no grant numbers for funding received from within IU. The funders had no role in study design, data collection and analysis, decision to publish, or preparation of the manuscript. The two authors Sarath Chandra Janga and Okiemute B. Omoru were supported by the R01GM123314 grant number from NIH.

**Competing interests:** The authors have declared that no competing interests exist.

While certain genera are non-pathogenic in Humans [5, 6], the genera of beta-coronaviruses comprise most human coronaviruses (HCoVs), including the SARS-CoV-1, MERS-CoV, HCoVOC43, HCoV-HKU1, and SARS-CoV-2 [7]. Beta-coronaviruses, including SARS-CoV-2, are highly pathogenic and are responsible for life-threatening respiratory infections in humans.

The SARS-CoV-2 genome is approximately 30'000 nucleotides long. The nucleotide content of the viral genome consists majorly of two large open reading frames (ORF1a and ORF1b) and structural proteins spike (S), envelope (E), membrane (M), and nucleocapsid (N) proteins, as well as several accessory proteins known as Open Reading Frames (ORF) 3a, 6, 7a, 7b, 8, and 10 [5, 8]. The structural proteins are responsible for viral assembly and suppressing the host's immune response [9, 10].

The first steps of coronavirus infection involve the viral entry into the host cell via binding of the Spike (S) protein to the cellular entry receptors for attachment to the receptor-binding site of the hosts cell membrane, fusion, and the release of the viral RNA into the cell. In humans, the host cellular receptor for SARS-CoV-2 is human angiotensin-converting enzyme 2 (ACE2) [11]. The interaction between Spike and ACE2 determines the viral response and pathogenicity [12–14]. After entry, SARS-CoV-2 expresses and replicates its genomic RNA to produce full-length copies, integrated into the newly created viral particles [11]. SARS-CoV-2's genome encodes NSPs, which are essential for viral RNA synthesis, and structural proteins necessary for virion assembly [15].

Coronavirus RNA-dependent RNA synthesis includes two differentiated processes of genome replication and transcription of a collection of sub-genomic RNAs. The sub-genomic RNAs encode the viral structural and accessory proteins. These RNAs are produced by discontinuous transcription where the synthesis of the negative-sense strand is disrupted. The resulting strand will then produce a plus RNA strand sub-genomic RNA. The complex replication/transcription machinery production of a series of sub-genomic RNAs through the process of template switching during negative-sense RNA synthesis [16, 17].

Beta-coronaviruses can form long-range high-order RNA-RNA interactions that contribute to template switch and consequently regulate the viral transcription and regulatory pathways for the production of sub-genomic mRNAs [16]. Long-range interactions are generally found in positive-strand viruses [18, 19]. The longest RNA-RNA interaction found so far spans ~26000 and is involved in a sub-genomic RNA synthesis in coronaviruses [18]. Mediated by stabilizing proteins, such interactions impact the tertiary structure of the genomic RNA, facilitating binding of the 5' UTR Transcript Regulatory Sequences (TRS) to the regulatory sequence upstream of a particular gene, leading to the switching of minus strand template to that of the gene's sub-genomic transcript. Regulation of the *N*-gene sub-genomic transcript is a fair example of such high-order RNA-RNA interactions [16]. Although some efforts have been made to investigate RNA-RNA interactions in in general of SARS-CoV-2 [20], It is very difficult to identify all the genomic RNA regions that are involved in such intricate interactions, presenting challenges to finding novel interacting regions within the virus [18].

The co-evolution of coronaviruses with their hosts is navigated by genetic variations made possible by its large genome [21], recombination frequency (of up to 25% for the entire genome in vivo) [22, 23], and a high mutation rate [24, 25]. SARS-CoV-2's mutation occurs spontaneously during replication. Thousands of aggregate mutations have occurred since the emergence of the virus [26]. A significant cause of concern about SARS-CoV-2's mutations is a change that could lead to a highly lethal infection or a failure on the effects of the current vaccines [27]. It is known that the strain with the highest similarity to SARS-CoV-2 is SARS-CoV-1. Similar to SARS-CoV-2, SARS-CoV-1 has a genome length of around 30kb (29'751nt), and its similarity ratio to the SARS-CoV-2 genome is 82.45% [28]. The genomic differences explain

the disparities in both viruses' dispersal and immune evasion [29]. The percentage similarity of the Spike protein of SARS-CoV-2 and SARS-CoV-1 is 97.71%. Spike protein's Receptor Binding Domain (RBD) which is the most variable part of the coronavirus genome [30], has 74.41% similarity. In fact, computational analysis has affirmed that the RBD sequence of SARS-CoV-2 differs from those observed to be ideal in SARS-CoV-1 [31]; hence, the high-affinity binding of the SARS-CoV-2 RBD to the human ACE2 is consequently due to natural selection on human ACE2, which allows for a solution for binding [32]. A significant difference between the Spike regions of both viruses is a polybasic insertion at the S1/S2 cleavage site, resulting from a 12-nt insert in the Spike region of SARS-CoV-2 that does not exist in SARS-CoV. In addition to increasing Spike protein infectivity, the 12-nt insert may also have a role on the RNA level, since it has a high GC content (CCUCGGCGGGCA; positions 23,603–23,614 of the reference). Similarity of other structural proteins are as follows: E-96%, M-89.41%, and N- 85.41%. The similarity between the structural protein of SARS-CoV-2 and other Coronaviruses is less than 50% [33].

RNA structures can play critical roles in the life cycle of Beta-coronaviruses. For instance, studies have reported that SARS-CoV-2's genomic RNA occupy some of the hosts MiRNAs that control immune regulated genes, thus depriving them of their function [34]. Recent studies have found locally stable RNA structures within the SARS-CoV-2 genome [35–38]. Moreover, *in-vivo* RNA structure prediction methods such as dimethyl sulfate mutational profiling with sequencing (DMS-MaP-seq) suggest that SARS-CoV-2 forms RNA structures within most of its genome [37], some of the possible relevance to the virus life cycle. These RNA structures can potentially be the target of RNA-based therapeutic applications [39, 40], or may lead to methods for inhibiting viral growth [41].

The Spike gene has been observed for having conserved RNA structural elements [42]. The 12-nt insert, which does not exist in Spike region of SARS-CoV-1, also contains unusually high GC composition, increasing its likelihood to have a role on the RNA level as well as protein level. In this work, we investigate the Spike gene on an RNA level. Using an *in-silico* fragment-based method, we compare the original SARS-CoV-2 sequence with its closest relative SAR-CoV-1 for any sign of major long-range RNA-RNA interactions that involve a genomic segment on the *Spike* region. The impact of locally stable RNA structures on the long-range predictions are also investigated. Subsequently, we considered the population of evolving SARS-CoV-2 sequences available worldwide to further investigate the conservation of our inferred interactions.

## Materials and methods

### Data

We used the SARS-CoV-2 isolate Wuhan-Hu-1 (NC_045512.2) and SARS-CoV-1 (NC_004718.2) reference sequences for identifying long-range RNA-RNA interactions in each of the viruses. For population-based sequence-covariance analyses, a set of 2,348,494 aligned full-length SARS-CoV-2 genome sequences were taken from the Nextstrain project [43] on December 9, 2021. The sequences were originally from the Global Initiative on Sharing All Influenza Data (GISAID) platform [44–46] (https://www.gisaid.org/) and were subsequently filtered for high quality sequence (nextstrain.org, filename: filtered.fasta.xz). We further filtered the sequences for having no ambiguous nucleotides in desired locations which resulted in a total of 2,068,427 sequences. Finally, we performed down-sampling to around 10 percent of original size (206,745 sequences) due to computational complexity constraints. S1 Table contains the corresponding GISAID accession numbers for the 206,745 sequences.

### Predicting RNA-RNA interactions

Genome-wide RNA-RNA interaction between the Spike region (query) and the genomic RNA of SARS-CoV-2 (target) were predicted using IntaRNA [47–50]. First, the Spike region was divided into smaller regions using a sliding window of length 500nt and overlap of 50nt. Each segment was then used as the query parameter by IntaRNA using search mode parameters (−mode H−outNumber 5−outOverlap Q). The parameters allowed for extracting top 5 non-overlapping targets on the full genome that form thermodynamically favorable RNA-RNA base-pairing interactions with a region on the corresponding query segment. Targets that were at least 1000nt apart from their query counterparts were subsequently kept. A similar procedure was carried out on SARS-CoV-1.

Different components of the RNAstructure software package [51] along with other tools were used for secondary structure predictions. Individual base-pair probabilities are according to McCaskill's partition function [52, 53].

### Compensatory mutations analysis of long-range RNA-RNA interactions

Compensatory mutations within the multiple sequence alignments were investigated using the R-scape software package [54–57], which analyzes covariation in nucleotide pairs in the population to infer possible compensatory mutations in an RNA base pair. If the consensus RNA secondary structure is not provided by the user, the software is also capable of predicting the consensus structure from the population of sequences using an implementation of the CaCo-Fold algorithm.

Compensatory (covarying) mutations for long-range RNA-RNA interactions were analyzed by retrieving the two sequence segments that constitute the desired RNA-RNA interaction for all downloaded SARS-CoV-2 sequences. Pairs of sequence segments were extended on each of their ends by 5nt (totaling 20nt) and concatenated. Then, the long-range RNA-RNA interacting structure was predicted by finding the consensus secondary structure within the population of sequences in the dataset using R-scape implementation of CaCoFold. The consensus structure was compared to bifold predictions for verification. Nucleotide pairs belonging to the consensus structure were then examined within the dataset for evidence of covariation using the built-in survival function that plots the distribution of base pairs with respect to their corresponding covariation scores.

## Results

Long-range RNA-RNA base-pairing interactions were predicted between the *Spike* region and the full genome for both SARS-CoV-1 and SARS-CoV-2 using IntaRNA software package (Fig 1). For each genome, the *Spike* region was extended 50nt on both directions. *Spike* sequence segments of length 500nt were analyzed separately for possible long-range interactions with their corresponding genomes (See Materials and Methods for details). We considered an arbitrary maximum of five hits (the optimal interaction and another four sub-optimal interactions) for each analysis. Fig 1 shows the location of all the hits in both the genomes.

Long-range RNA-RNA predictions between *Spike* and the full genome are spread across almost all other genes for both SARS-CoV-1 and SARS-CoV-2 genomes. These interactions consisted of different thermodynamic stabilities and included interacting regions of as short as around 20nt. S2 Table contains details about each hit. There were some major observations in our comparison. First, no interacting candidate was observed between the *Spike* and *E* genes for neither of the stains. Second, unlike SARS-CoV-2, a considerably long segment on SARS-CoV-1 *Spike* gene did not contain any prediction with the rest of the genome. In fact, the query segment of SARS-CoV-1 *Spike* (23,238–23,737) contained only two hits, while other

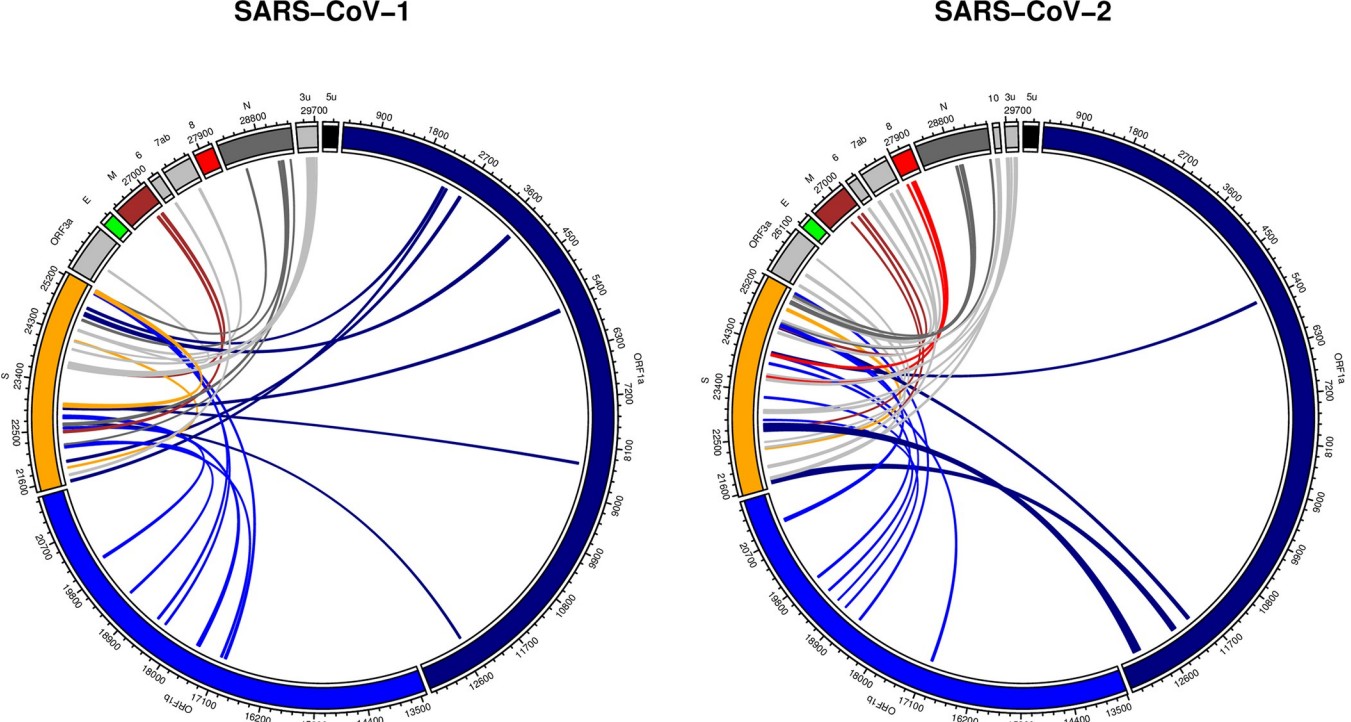

**Fig 1. Predicted long-range RNA-RNA base-pairing interactions between *Spike* and the full genomic RNA.** *Spike* sequence segments of length 500nt and overlap of 50nt were queried against the full genomes using IntaRNA software package. Each individual test resulted in at most five hits. All hits are summarized for both SARS-CoV-1 and SARS-CoV-2 (See Materials and Methods for details).

segments (on both strains) resulted at least four long-range predictions. The no-hit region corresponded to (23238–23698) on SARS-CoV-1, in specific. Finally, no prediction was observed between the SARS-CoV-1 *Spike* and *ORF8* regions, while this was not true for SARS-CoV-2. As we can see in Fig 1, there are multiple hits between *Spike* and *ORF8* for SARS-CoV-2.

There was a total of 69 long-range interactions across both viruses. Table 1 summarizes the top quantile hits. The ranking of interactions was based on using their residual values against a generalized linear model that estimates interaction energy from interaction length. The reason for choice of model was that the fact that expected interaction energy is related to sequence length. The built-in function glm(energy~length, data = data, family = "gaussian") in R programming language was used to fit the model. Length was a significant factor in the model with (Pr(>|t|) for length = 0.00067 which confirmed our assumption about impact of length on interaction energy (See Table 1 caption for Model details). Residual values were used to rank the interactions, since hits with lower residuals imply higher stability compared to other hits.

Focusing only on SARS-CoV-2 hits, the top hit corresponds to the beginning of the *Spike* gene. In fact, the interaction overlaps with the upstream region of *Spike*. Interestingly, the second and third top hits are exactly adjacent to each other on the *Spike* region. Target regions shown in rank 3 and rank 7 are (24114–24157 *Spike*) and (24084–24114 *Spike*) and interact with their corresponding regions on ORF1a and ORF1b, respectively. Base-pair level interaction details for the top three interactions can be found in S1 Fig. From amongst the predicted interactions, we decided to focus on further investigating the major hit between *Spike* and *ORF8* of SARS-CoV-2. This rather qualitative choice was based on the following: As mentioned before, we were primarily interested in novel interactions and ORF8 was not observed

**Table 1. Top quantile predicted long-range RNA-RNA base-pairing interactions between the *Spike* region the full genome for both SARS-CoV-1 and SARS-CoV-2 using IntaRNA software package.**

| Rank | SARS-CoV | Hit Start | Hit End | Target Start | Target End | Total Length | Energy | Residual | Target Gene |
|------|----------|-----------|---------|--------------|------------|--------------|--------|----------|-------------|
| **1** | **2** | **21639** | **21750** | **12261** | **12355** | **207** | **-26.29** | **-7.3397887** | **ORF1a** |
| 2 | 1 | 22604 | 22631 | 12507 | 12532 | 54 | -21.23 | -7.1602061 | ORF1a |
| **3** | **2** | **24114** | **24157** | **5367** | **5402** | **80** | **-19.93** | **-5.0308541** | **ORF1a** |
| 4 | 1 | 24396 | 24414 | 25582 | 25602 | 40 | -18.19 | -4.5667802 | ORF3a |
| 5 | 1 | 24841 | 24877 | 2247 | 2288 | 79 | -19.26 | -4.3927522 | ORF1a |
| 6 | 1 | 25198 | 25239 | 17014 | 17053 | 82 | -19.1 | -4.1370578 | ORF1b |
| **7** | **2** | **24084** | **24114** | **17012** | **17046** | **66** | **-18.4** | **-3.9474282** | **ORF1b** |
| 8 | 1 | 23698 | 23734 | 26957 | 27000 | 81 | -18.87 | -3.9389559 | M |
| **9** | **2** | **23271** | **23295** | **19401** | **19423** | **48** | **-17.4** | **-3.521595** | **ORF1b** |
| **10** | **2** | **22846** | **22862** | **18954** | **18970** | **34** | **-16.92** | **-3.4881691** | **ORF1b** |
| **11*** | **2** | **23660** | **23703** | **28025** | **28060** | **80** | **-18.07** | **-3.1708541** | **ORF8** |
| **12** | **2** | **22303** | **22337** | **24984** | **25023** | **75** | **-17.79** | **-3.0503449** | **Spike** |
| **13** | **2** | **24984** | **25023** | **22303** | **22337** | **75** | **-17.79** | **-3.0503449** | **Spike** |
| 14 | 1 | 21523 | 21559 | 2321 | 2354 | 71 | -17.53 | -2.9179375 | ORF1a |
| **15** | **2** | **25331** | **25358** | **18595** | **18620** | **54** | **-16.79** | **-2.7202061** | **ORF1b** |
| 16 | 1 | 24667 | 24677 | 24104 | 24114 | 22 | -15.37 | -2.320947 | Spike |
| **17** | **2** | **24648** | **24660** | **19153** | **19165** | **26** | **-15.44** | **-2.2633543** | **ORF1b** |
| 18 | 1 | 22530 | 22558 | 20039 | 20077 | 68 | -16.67 | -2.1536319 | ORF1b |

See Materials and Methods for details. There was a total of 69 independent hits across both genomes. Complete results included as S2 Table. Column `SARS-CoV` denotes the strain. Column `TotalLength` denotes length of the interacting regions (query + target). Ranking is according to residual values against the generalized linear model where length of interaction was used to estimate interaction energy. The built-in function `glm(energy~length, data = data, family = "gaussian")` in R programming language was used to fit the model. Length coefficient = -0.03190. Length was a significant factor in the model. (Pr(>|t|) for length = 0.00067. Median of residuals = -0.2287). 1-Quantile of residuals = -2.1536. SARS-CoV-2 hits are shown as bold. Rank 11 also shown with * denotes the SARS-CoV-2 Spike-ORF8 interaction.

to contain any long-range integration in SARS-CoV-1. In addition, as will be explained in the next section, the above hit is within the top quantile predictions (Table 1) and is not sensitive to the top-5-hit choice of cut-off (data not shown).

### *Spike-ORF8* RNA-RNA interaction

The interaction between *Spike* and *ORF8* with the highest ranking appears as the 11[th] top hit within a total of 69, under a generalized linear model that estimates interaction energy from sum of lengths of interacting sequences. It is also the 6[th] top hit within SARS-CoV-2. Base-pairing interactions between SARS-CoV-2 *Spike* and *ORF8* are shown in Fig 2. Intervals (23660–23703 *Spike*) and (28025–28060 *ORF8*) consist of a total of 80nt and have a stabilizing energy of -18.07 kcal/Mol. Fig 2 shows the individual base pairs of the above hit, denoted here as *Spike-ORF8 interaction*. Pairs shown by '+' symbol point to stable sub-interactions and thus likely to be starting points of the full long-range RNA-RNA interaction (predictions according to IntaRNA). This sub-interaction is shown within the red rectangle in Fig 2 and denoted as the *Core* interacting region.

The predicted Spike-ORF8 interaction was analyzed for compensatory mutations. Sequence segments were extended 5nt to avoid unwanted base-pairing in the consensus structure prediction. Resulting intervals were (23655–23708 *Spike*) and (28020–28065 *ORF8*). A total of 206,745 sequence segments each corresponding to a particular viral sequence was used for the analysis. Sequences were a down-sampled selection of nearly two million SARS-CoV-2

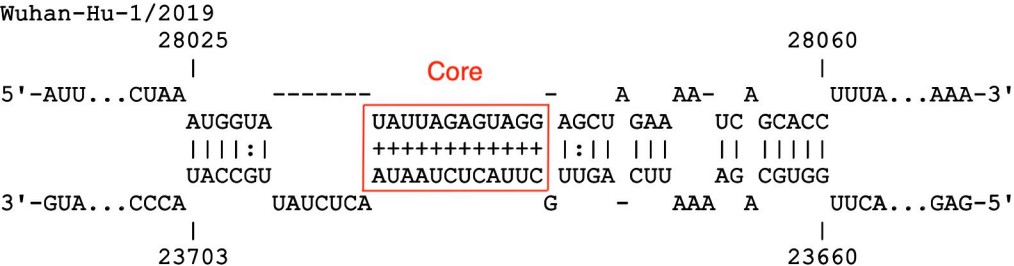

**Fig 2. Long-range RNA-RNA interaction between *Spike* and *ORF8* regions of SARS-CoV-2 genome.** Interacting intervals are (23660–23703 *Spike*) and (28025–28060 *ORF8*). Prediction done via IntraRNA software. Base pairs with 'plus' notation denote stable sub-interactions. The stable sub-interaction is shown within the red rectangle in and denoted as the *Core* interacting region: (23679–23690 *Spike*) and (28031–28042 *ORF8*).

sequences (See Materials and Methods for detail). No significantly covarying mutations were detected by R-scape. Table 2 shows the coordinates of all base pairs for which variation was observed. Column `power` an output of the R-scape software, denotes the statistical power of substitutions.

Interestingly, comparing Fig 1 and Table 2, we can see that the base pairings within the *Core* interacting region also have lower variation in the population of sequences than other predicted base pairs of the interaction. They are shown in Table 2 in black bold.

Fig 3 illustrates the individual pairing configurations within the RNA-RNA interaction. Results were according to the consensus structure prediction algorithm CaCoFold built in the R-scape software. Both structure prediction methods IntaRNA (thermodynamic long-range) and CaCoFold (consensus structure) had consistent results in predicting most based pairs including those in the *Core* integrating region. The first part of the interaction, however, is predicted by intaRNA but not CaCoFold. The coordinate for this region is (23698–23703 *Spike*) and (28025–28030 *ORF8*). Four base pairs with highest number of mutations are shown in bold black in Fig 3. Nucleotide position with highest observed mutation was G28048U *ORF8*, with 36,366 occurrences in a total of 206,745 viral sequences. This mutation does not support the predicted interaction. Mutation C23664U *Spike* was observed 2207 times and was the second highest mutation observed. This mutation accommodates for the Spike-ORF8 interaction. Adjacent to this base pair, mutation G28045U *ORF8* with frequency 329 also accommodates for the interaction stability. The fourth most frequent mutation was C28045U *ORF8*. It was observed 329 times which also accommodated the predicted Spike-ORF8 interaction. Base pairs falling within the *Core* interacting region are shown cells shaded red in Table 2. Sequence variation in almost all these base pairs is zero. The flanking sequences on both ends of interactions did not form any base pairing with each other as expected by IntaRNA results.

## Local RNA analysis in *Spike*

The local stability of RNA structure in the vicinity of the (23660–23703 *Spike*) was evaluated and compared to its SARS-CoV-1 counterpart. The original interval was extended by 100nt on both directions on the SARS-CoV-2 genome, resulting region (23560–23803 *Spike*). The region that aligned with the above selection on SARS-CoV-1 was selected for comparison, (23447–23650 *Spike* S2 and S3 Figs show the base pair probabilities for both SARS-CoV-2 (23560–23803 *Spike*) and its corresponding region in SARS-CoV-1 (23447–23650 *Spike*). Base

**Table 2. Coordinates of interacting base pairs between (23660–23703 *Spike*) and (28025–28060 *ORF8*) for which nucleotide variations were observed.**

| *Spike* | *ORF8* | Power |
|---------|--------|-------|
| 23660 | 28060 | 0 |
| 23661 | 28059 | 0 |
| 23662 | 28058 | 0 |
| **23663** | **28057** | **0.08** |
| **23664** | **28056** | **0.11** |
| 23671 | 28050 | 0 |
| 23672 | 28049 | 0 |
| **23673** | **28048** | **0.39** |
| 23674 | 28046 | 0 |
| **23675** | **28045** | **0.05** |
| 23676 | 28044 | 0.04 |
| 23677 | 28043 | 0 |
| **23679** | **28042** | **0** |
| **23680** | **28041** | **0.01** |
| **23681** | **28040** | **0** |
| **23682** | **28039** | **0** |
| **23683** | **28038** | **0** |
| **23684** | **28037** | **0** |
| **23685** | **28036** | **0** |
| **23686** | **28035** | **0** |
| **23687** | **28034** | **0** |
| **23688** | **28033** | **0.01** |
| **23689** | **28032** | **0** |
| **23690** | **28031** | **0** |

Total number of sequences was 206,745. Column `power` is an output of the R-scape software that is proportional to the statistical power of substitutions. Mutations in coordinates in black bold are shown Fig 3. Base pairs within the *Core* interacting region (Fig 2) are shown in cells shaded red.

pairs colored in red are those with higher likelihood of forming. As we can see, there are major differences in the base-pairing probability patterns between the two sequences. The black bar shows the approximate location of the Spike-ORF8 interaction. As we can see this location seems to contain many base pairs that can form local base pairs. The corresponding location on SARS-CoV-1, for which no long-range interaction with *ORF8* was observed, seems to have relatively less locally stable bases pairs (comparing red base pairs between S2 and S3 Figs). This observation was also true for another arbitrary selection of sequence segments. Overall, region of *Spike* that is predicted to base pair with *ORF8*, also tends to form a local structure which seems to be mutually exclusive from the *ORF8* interaction.

## Discussion

The Spike region of SARS-CoV-2 RNA was investigated for novel genomic long-range RNA-RNA interaction. Fragment-based *in-silico* predictions were performed on the reference sequence and compared to those for the reference sequence of SARS-CoV-1 that was responsible for the 2002–2003 outbreak.

The predictions were inclusive and made in favor of more sub-optimal but diverse results. They provide a collection of top non-overlapping candidate regions on the reference sequences

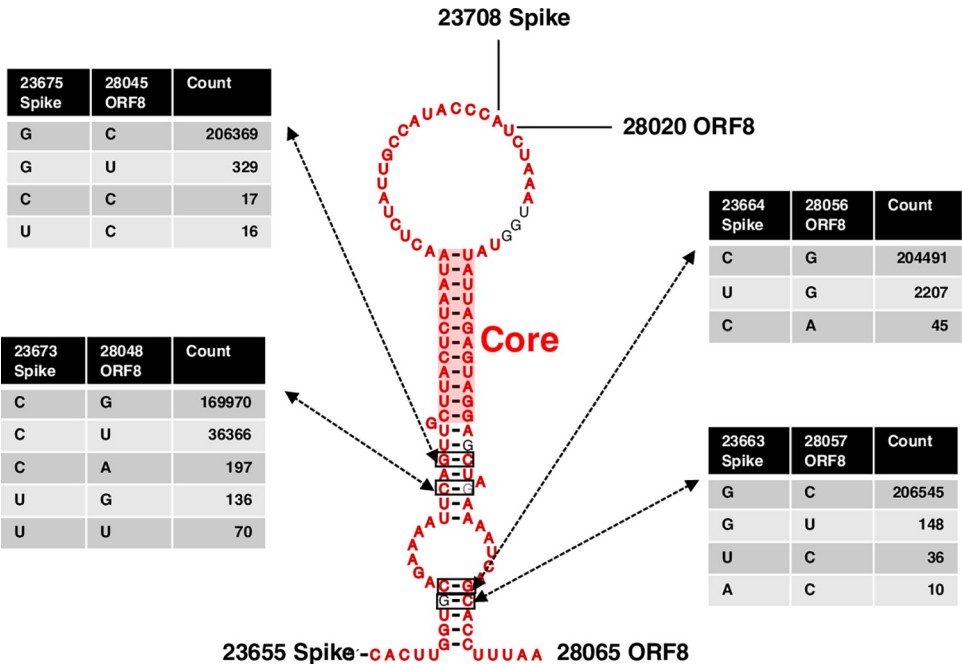

**Fig 3. Consensus structure of the predicted Spike-ORF8 RNA-RNA interaction.** RNA-RNA interaction coordinates were (23660–23703 *Spike*) and (28025–28060 *ORF8*). Total number of sequences was 206,745. Number of mutations observed for four locations with highest power are shown. The *Core* interacting region is shown by the transparent red rectangle.

that can potentially form thermodynamically favorable RNA-RNA base pairing with a sub-region on their corresponding *Spike* (Fig 1). We found RNA structural differences between corresponding regions in SARS-CoV-2 and SARS-CoV-1. It is worth noting, however, that the cut-off for storing number of interactions was chosen arbitrary, which implies there may be more predictions that are not included in Fig 1.

Top interacting regions were ranked according to their relative thermodynamic stabilities with regards to total length of interaction, using a generalized linear model. Table 2 shows the top quantile of results (See S2 Table for full results). Some of the predictions are as follows. Strongest interactions that occurred on the SARS-CoV-2 genomic RNA were between *Spike* and *ORF1ab*. A region in the beginning of SARS-CoV-2 Spike (21639–21750) formed an interaction with a region on *ORF1ab* (12261–12355) with a predicted free energy of -26.29 kcal/Mol, highest amongst both viruses. Details about base pairing interactions of the top three hits is presented in S1 Fig. The second and third strongest predictions on SARS-CoV-2 also occurred on ORF1ab but formed continuous region on the side of *Spike*. Regions (24114–24157 *Spike*) and (24084–24114 *Spike*) intersect at position 24114 but interact with distant regions on *ORF1ab*, namely the beginning (5367–5402) and the middle (17012–17046), respectively. This observation was unique, since interactions were allowed to overlap on *Spike* by the IntaRNA software, but they only have one nucleotide overlap on *Spike* (See Table 2, rows 3 and 7 for details). Some of the other interesting observations were the fact that a significantly long region of SARS-CoV-1 *Spike* (23238–23698), around 460nt, did not form any long-range RNA-RNA interacting predictions with any part of the genome, despite the software's flexibility to allow for sub-optimal hits. This lack of predictions was not observed in SARS-CoV-2 *Spike*.

Most genes and annotated regions contained several interacting regions with the Spike gene in both reference genomes SARS-CoV-1 and SARS-CoV-2 (comparing Fig 1A and 1B). The ranking of strength of base-pairings, however, were dramatically different between corresponding genes. For instance, the strongest ranked interaction between *Spike* and *M* in SARS-CoV-1 was 8[th] while this number dropped to 40 for SARS-CoV-2 (See S2 Table). The only gene that did not contain any predictions was the *E* gene. No thermodynamically stable interacting candidate was observed on neither of SARS-CoV-1 and SARS-CoV-2 reference genomes.

SARS-CoV-2 contained a few regions that can potentially form long-range RNA-RNA interactions with the *Spike* and *ORF8* regions on different locations (Fig 1B, red links), while SARS-CoV-1 didn't contain any. The ranking of the highest observed interaction stability fell within the first quantile of results (ranking 11). Regions (23660–23703 *Spike*) and (28025–28060 *ORF8*) formed an RNA-RNA interaction with free energy of -18.07 kcal/Mol. While the other results were interesting and worth further investigation, our focus was further analysis of the above Spike-ORF8 interaction, due to the strong gene-based observed contrast between the SARS-CoV-1 and SARS-CoV-2. A sub-interacting region (23679–23690 *Spike*) and (28031–28042 *ORF8*) within the above interval was predicted to have a higher likelihood to form thermodynamic stable base pairings, denoted here as the *Core* integrating region (Fig 2).

The population of SARS-CoV-2 sequences were analyzed for signs of sequence co-variation that might validate the above S-ORF8 RNA-RNA base-pairing interaction. From amongst the nearly 20 million sequences, 206,745 (roughly 10%) were randomly selected for the analysis, due to limitations in computational complexity. The aligned SARS-CoV-2 sequences were investigated for compensatory mutations that might occur within and between *Spike-ORF8* binding location. Although not any significantly covarying mutations were observed, the positions of polymorphisms were in support of the *in-silico* results. Interestingly the *Core* interacting region was observed to tolerate less mutations (location shown in bold red, Table 2). The lower variance in the more stable base pairs is in support of the Spike-ORF8 RNA-RNA interaction. Other regions of the interaction either had higher variation or did not even appear in the consensus structure predicted by CaCoFold. The integration of thermodynamic-based predictions and sequence variation identify a region (*Core* region) for the predicted Spike-ORF8 RNA-RNA interaction.

Observed mutations within the interacting region, however, had conflicting implications, with some such as C28045U *ORF8*, G28048U *ORF8*, C23664U *Spike* being in favor of the interactions and some such as G28048U not accommodating for base pairing (Fig 3). Being a synonymous mutation, C28045U has been previously identified as one of the polymorphic positions of *ORF8* [58]. In the mentioned work, in the local RNA secondary structure prediction of *ORF8*, C28045U is unpaired, while in the predicted long-range RNA-RNA interaction with *Spike*, it pairs with G23675. The C28045U variation, hence, is suggestive of the long-range Spike-ORF8 interaction. Further investigation on the above set of mutations along the evolutionary trajectory of the virus is needed for a more comprehensive conclusion about their possible roles. In addition, since the data was filtered and aligned for having no long inspersions, deletions, or ambiguous nucleotides, certain meaningful sequence variations might not have been accounted for in the analysis.

Local RNA structure analyses on the *Spike* region suggests an increase in locally stable RNA structures in the vicinity of the Spike-ORF8 interaction. There is a conserved RNA stem-loop, namely S1, which has been previously found in SAR-CoV-1 sequences [42]. This stem is roughly 30nt upstream of the Spike-ORF8 interaction and its stability was confirmed by different *in-silico* programs in both SARS-CoV-1 and SAR-CoV-2 sequences. Immediately upstream of the conserved stem, there is the high-GC content 12-nt insert in the *Spike* region

(23603–23614), which is present in SARS-CoV-2 but absent in SARS-CoV-1. The insert is roughly 50nt upstream of the predicted Spike-ORF8 interaction. Given the above comparisons to SARS-CoV-1, it seems that this region of *Spike* is undergoing local RNA structural changes as well as having affinity to form a long-range interaction with ORF8.

Locally stable RNA base pairs and the long-range Spike-ORF8 base-pairing interactions are mutually exclusive. Base pair probability distributions of corresponding regions on *Spike* in both SARS-CoV-1 and SARS-CoV-2 reveal that the same nucleotides that can pair with ORF8, are also likely to form local base pairs within *Spike* (Fig 3, red base pairs falling under the bar). Ironically, the corresponding region on SARS-CoV-1, for which there were no signs of long-range interaction with ORF8, is observed to have less deterministic local base-pairing probabilities (Comparing S2 and S3 Figs, range indicated by black bar). One possibility is that a complex RNA structure may be emerging within the specified region of *Spike* in SARS-CoV-2 that can form RNA-RNA interaction with ORF8, at certain times can avoid the interaction at others. Whether the predicted long-range Spike-ORF8 interaction is in competition or cooperation with other local elements of *Spike* such as the 12-nt polybasic insert in SARS-CoV-2, is subject to speculation about *in-vivo* conformational specifics.

Given our methodology, it cannot be inferred if the predicted *Spike-ORF8* RNA-RNA interaction could form in the genomic RNA or within a sub-genomic RNA, or even between two different sub-genomic RNAs, since only small fragments of sequences were effectively considered in our predictions. An interesting possibility is the genomic scenario where the hypothesized interaction can potentially impact template switch during negative strand synthesis. Template switch in Beta-coronaviruses might occur if the TSR element downstream of the 5'UTR is in proximity of the TSR element immediately upstream of a viral gene. Such complex genomic conformation may involve other RNA-RNA as mediators. The dE-pE (Fig 2 of [16]) acts as such mediator RNA binding locations to facilitate a discontinuous negative strand synthesis of the viral genome, leading to N-gene sub-genomic RNA. The coronavirus nucleocapsid (N) is known to be a structural protein that forms complexes with genomic RNA, interacts with the viral membrane protein during virion assembly and plays a critical role in enhancing the efficiency of virus transcription and assembly [16]. The predicted Spike-ORF8 interaction here is 200nt upstream of the N-gene TSR [58]. Although high-order RNA-RNA interactions needed for template switch can be more complex and may involve the 5'UTR as well, the predicted Spike-ORF8 interaction could indeed be acting as an additional mediator step to bring the TRS elements of 5'UTR and the coronavirus *N*-gene closer to each other. It could be speculated that the Spike-ORF8 interaction is taking part in regulating sub-genomic RNA production. Since the first gene downstream of Spike-ORF8 interaction happens to be the *N*-gene, the binding location might be affecting the *N*-gene sub-genomic RNA production.

Amongst coronaviruses, *ORF8* is a rapidly evolving hypervariable gene that undergoes deletions to possibly adapt to human host [58–61]. It has also been previously observed that patients infected with SARS-CoV-2 variants with a 382-nucleotide deletion (Δ382) in *ORF8* had milder symptoms [62]. In addition, *ORF8* contains RNA structural features [58]. While this observation may very well be due to impact of absence of the translated protein, *ORF8* RNA structural characteristics of the genome may also play a role in the viral life cycle, making long-range RNA-RNA prediction with *Spike* a less remote possibility. A comprehensive exploration of the predicted Spike-ORF8 interaction amongst SARS-CoV-2 variants and evaluating corresponding sub-genomic RNA production rates of these variants may lead to further clues about the predicted long-range Spike-ORF8 RNA-RNA interaction, which can be rewarding for therapeutic purposes.

## Supporting information

**S1 Fig. Top three interacting regions with SARS-CoV-2 Spike.** Corresponding ranking of the hits are also included. Generalized linear model was used to rank hits with highest interaction energy relative to interaction length (Table 1).
(TIF)

**S2 Fig. Base pair probabilities for aligned segments of SARS-CoV-1 *Spike*.** Probabilities were calculated using McCaskill's partition function [51, 52]. Coordinates: SARS-CoV-1 (23447–23650 *Spike*). SARS-CoV-2 Spike/ORF8 region (23660–23703) was mapped on SARS-CoV-1, with coordinates: SARS-CoV-1 (23534–23575 *Spike*) and extended on both directions by 100nt. The resulting region is highlighted as a black bar.
(TIF)

**S3 Fig. Base pair probabilities for aligned segments of SARS-CoV-2 *Spike*.** Probabilities were calculated using McCaskill's partition function [51, 52]. Coordinates: SARS-CoV-2 (23560–23803 *Spike*). The Spike/ORF8 region (23660–23703), shown as a black bar was extended on both directions by 100nt.
(TIF)

**S1 Table. GISAID accession numbers.** Accession numbers of the 206,745 SARS-CoV-2 sequences used in the study.
(TXT)

**S2 Table. Ranking of predicted RNA-RNA base-pairing interactions.** Predicted long-range RNA-RNA base-pairing interactions between the *Spike* region the full genome for both SARS-CoV-1 and SARS-CoV-2 using IntaRNA software package. See Materials and Methods for details. There was a total of 69 independent hits across both genomes. Column SARS-CoV denotes the strain. Column TotalLength denotes length of the interacting regions (query + target). Ranking is according to residual values against the generalized linear model where length of interaction was used to estimate interaction energy. The built-in function glm(energy~length, data = data, family = "gaussian")in R programming language was used to fit the model. Length coefficient = -0.03190. Length was a significant factor in the model. ($Pr(>|t|)$ for length = 0.00067. Median of residuals = -0.2287). 1-Quantile of residuals = -2.1536. SARS-CoV-2 hits are shown as bold. Rank 11 also shown with $^*$ denotes the SARS-CoV-2 Spike-ORF8 interaction.
(CSV)

## Author Contributions

**Conceptualization:** Filipe Pereira, Amirhossein Manzourolajdad.

**Data curation:** Okiemute Beatrice Omoru, Amirhossein Manzourolajdad.

**Formal analysis:** Okiemute Beatrice Omoru, Amirhossein Manzourolajdad.

**Funding acquisition:** Sarath Chandra Janga.

**Investigation:** Filipe Pereira, Sarath Chandra Janga, Amirhossein Manzourolajdad.

**Methodology:** Filipe Pereira, Amirhossein Manzourolajdad.

**Project administration:** Sarath Chandra Janga.

**Supervision:** Filipe Pereira, Sarath Chandra Janga.

**Visualization:** Okiemute Beatrice Omoru.

**Writing – original draft:** Okiemute Beatrice Omoru, Filipe Pereira, Amirhossein Manzourolajdad.

**Writing – review & editing:** Filipe Pereira, Sarath Chandra Janga.

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
