## [Decision Letter · Decision Letter 0]

13 Dec 2021

PONE-D-21-35236Evidence for a long-r ange RNA-RNA interaction between ORF8 and the downstream region of the Spike polybasic insertion of SARS-CoV-2PLOS ONE

Dear Dr. Manzourolajdad,

Thank you for submitting your manuscript to PLOS ONE. After careful consideration, we feel that it has merit but does not fully meet PLOS ONE’s publication criteria as it currently stands. Therefore, we invite you to submit a revised version of the manuscript that addresses the points raised during the review process.

The revised manuscript should address all the critical points raised by all reviewers.

We look forward to receiving your revised manuscript.

Kind regards,

Danny Barash

Academic Editor

PLOS ONE

Journal Requirements:

Reviewers' comments:

Reviewer's Responses to Questions

**Comments to the Author**

1. Is the manuscript technically sound, and do the data support the conclusions?

Reviewer #1: Partly

Reviewer #2: Yes

2. Has the statistical analysis been performed appropriately and rigorously? 

Reviewer #1: I Don't Know

Reviewer #2: Yes

3. Have the authors made all data underlying the findings in their manuscript fully available?

Reviewer #1: Yes

Reviewer #2: Yes

4. Is the manuscript presented in an intelligible fashion and written in standard English?

Reviewer #1: Yes

Reviewer #2: Yes

5. Review Comments to the Author

Reviewer #1: # Review for PONE-D-21-35236

Evidence for a long-r ange RNA-RNA interaction between ORF8 and the downstream region of the Spike polybasic insertion of SARS-CoV-2

Filipe Pereira and Amirhossein Manzourolajdad

# Summary

The article summarizes the investigation of potential long-range RNA-RNA interactions (RRIs) within the SARS-CoV-2 genome and related sequences in SARS-CoV-1. In details, the study focuses on the genomic subregion 23600-24200 that is part of the Spike gene. The region is defined by the intra-molecular helix formed by a SARS-CoV-2-specific 12nt insert located at the beginning of the region that forms stable base pairing with the region's end.

The authors performed IntaRNA-based RRI prediction to screen for RRIs that interlink the genomic subregion with other parts of the genome respectively for both viruses. Two high-scoring interactions were identified, i.e. a stable and conserved RRI of the region's end with ORF1ab and a non-conserved RRI that interlinks in SARS-CoV-2 the beginning of the region with ORF8.

Following the headlines of the Results section, the main findings by the authors are (1) locally stable structure within the region (2) the two stable interactions mentioned above and (3) support for the formation of the Spike-ORF8 RRI via covariation analyses.

# General remarks

I have couple of central issues with this manuscript in its current form.

## (1) 12 nt insert irrelevant and obscures the focus and informative value

Beside the definition of the genomic subregion of interest, the 12 nt insert is eventually unimportant for any statement supported by the study (beside observing that S1 stem loop is stable with and without the insert).

The authors write themselves "The contribution of the 12-nt insert [...] is still unclear".

Thus, I have to ask: why then using the 12-nt fragment as such a rigid anchor for the study? The authors use about 2/3 of the Spike gene somewhat at random (based on the 12nt and its structure formation) instead of the whole gene! The rational for taking the whole could be the same, i.e. "12-nt insertion is inside", but now the SARS-CoV-1 subregion is clearly defined (as the related full gene sequence) and not based on the mapping of a substructure region...

Using just a subregion of the gene

(a) causes problems with local structure prediction AT LEAST at the ends of the regions if not overall,

(b) has thus implications on RRI prediction, since IntaRNA incorporates the accessibility of interacting subregions, which is based on local structure prediction, and

(c) restricts the insights of the study to a hypthetical local fragment rather than a valid mRNA-like molecule.

## (2) mfe structures based on global folding are no local structure signal

Using minimum free energy (mfe) estimates based on GLOBAL structure prediction (i.e. allowing base pairing over the range of the RNA) and respective structure plots to discuss local structure formation is inappropriate. The mfe is only ONE possible stable structure and strongly depends on the underlying thermodynamic model. Mfe prediction gets decreasingly reliable (in its details) the longer the RNA. Thus, instead of individual base pairing (of the presented mfe structure), LOCAL base pair probabilities (eg visualized via dot plots) are a much sounder tool to investigate local structure formation of RNAs or genomic regions. And if the details of base pairing of of no importance, even local unpaired probabilities (related to the accessibility values used by IntaRNA), are simple tool to identify locally (un)structured regions etc. without detailing base pairing.

When presenting mfe predictions of different tools (Fig-1) that are based more or less on the same thermodynamic energy model, no further insight is gained...

Furthermore, structure representations (Fig-1, Fig-4) are

-) so small that they are not readable in print

-) due to the latter, coloring is completely lost in print

-) eventually unimportant and not discussed in details for the RRI prediction

## (3) There is no covariation as far as I can see...

Please check Fig-1a of

https://academic.oup.com/bioinformatics/article/36/10/3072/5729989

the only thing I observe in the provided data (Fig-3) is VARIATION, i.e. only one side of the base pair shows variants...

The whole covariation part has several flaws..

(a) the authors use the concatenated interacting subsequences as an input to the CO-FOLDING BASED R-scape software... This is terribly wrong, since no LINKER sequence spacing the two FRAGMENTS is used. Without it, the helix close to the of the interaction that is close to the point of concatenation cannot be formed due to STERICAL reasons respected by the structure prediction software... Thus, the co-variation study misses eventually 5 base pairs and even more important, the Spike region is ONE TOO SHORT! missing the final interacting U...

?!?

If done that way, ie. using the co-folding based approach, one has to insert a linker region of at least 3-5 nt, eg. poly-A or if possible poly-N, to allow for enough flexibility that the concatenated ends of the sequences can form a structure. Or why not extending the subregion range by 5nt in both directions? would be simple and avoid this central error..

(b) who does the counts in Fig-3b relate to the substitution count in Table-1? The authors refer to the manual of R-scape, which is in no way a proper scientific explanation, even more so since these counts are central for selecting pairs and position within the argumentation of the authors..

(c) the substitution counts incorporate AMBIGOUS nucleotides (Fig-3b)... ?!? What about sequencing errors etc. as a source of such ambiguity? For other parts of the study such subsequences were excluded, why not here?

(d) the authors MISS an important point: the IntaRNA predictions are annotated with based pairs that are likely to be formed first, the seed base pairs represented by "+" within the plot. These base pairs seem to correlate with LOW subsitution scores, which I would find a supportive and good sign! That is low sequence variation in the important sub-RRI!

Thus, an investigation and relation of VARIATION with RRI formation would be more appropriate than a co-variation study that shows there is no co-variation..

## (4) Why showing bi-/tri-molecule co-folding results?

I find no rational for nor any additional insights from the bi-/tri-molecule folding part of the manuscript. The description of the assembly of the tri-molecule prediction from bi-molecule graphs is very vague.

Maybe I missed it, but what's the point? Fig-4a shows the same RRI patterns as Fig-2a. The lack of structure relation of Fig-4b and Fig-2b can/will have multiple possible reasons that are not discussed at all..

If you are interested in co-formation of multiple RRIs, why not using a constraint IntaRNA prediction? Therein you can mark regions that are involved in base pairing (eg. an RRI) and you will get corrected predictions for the rest of the molecule EXCLUDING the ballast of local intra-molecular mfe structure (see (2)).

But eventually, that wasnt even the point, or was it? I am confused by this part..

## (5) No suboptimal RRIs investigated/shown/discussed ...

For RRIs, the same holds as discussed for (2). While base pairing probability prediction is much harder for RRI prediction compared to intra-molecular structure prediction, one can relate to suboptimal interactions as a substitute to assess the structural variety. This is needed at multiple parts within this manuscript:

(a) the extremely long UTF interaction in Fig-2b is most likely an artifact of the summation-based prediction scheme. Simply spoken: the more base pairs the better until accessibility penalties overweight the gain.

BUT: did you check suboptimal interactions of the region? I would expect much shorter sub-RRIs with similar stability (i.e. energy).

(b) Furthermore did you check if a similar ORF8 interaction is among the suboptimals of the SARS-CoV-1 predictions? Since the ORF8 SARS-CoV-2 RRI shows a lower energy that the UTF-RRI, the chance is high that it is "hidden" there...

# Conclusion

Given the length of my remarks and the bulk of flaws, I cannot recommend the manuscript in its current form for publication.

I hope the authors take the time to really reshape and rework this study before resubmitting as I took the time to point out the weaknesses.

Reviewer #2: Prior studies have discovered many long-range RNA-RNA interactions within the genomes of positive-strand RNA viruses, which have functional roles in fundamental viral processes. In this submission, Pereira and Manzourolajdad analyze SARS-CoV-2 genomic sequences from the GISAID database. They report a long-range interaction of a region containing a 12-nt insertion in the Spike protein region that is not present in SARS-CoV. They use standard methods to analyze RNA sequence data, and find locally stable structural features downstream of the insertion region, evidence for potentially functional interactions with ORF8 region, and some evidence for sequence covariation in the interacting regions. These findings can be useful to understand the biology of SARS-CoV-2 and develop therapies. Eventually, experimental data will be needed to ascertain the presence of these interactions in vivo and its functional significance. However, that may be beyond the scope of the present study. I have focused my comments on the computational analysis, which I hope the authors will consider addressing.

1. The 12 nt insertion site is 23,603-23,614 of the genome. The authors searched for long range interactions of any SARS-CoV-2 region with the region 23,600-24,107 and 23,917-24,118, i.e., only downstream region of the insertion site. Shouldn't equally long stretches upstream of the insertion site be considered as well? It is not clear to me why the present study has focused on the downstream region only.

2. Similar to my previous question, it is not clear why the overlapping regions 23,600-24,107 and 23,917-24,118 were analyzed separately. It will be helpful if the authors can comment on this.

3. The developers of IntaRNA have shown that integrating experimentally obtained chemical probing data with IntaRNA can significantly improve RNA-RNA interaction prediction. Since such data is available for SARS-CoV-2, have the authors considered using it in their analysis?

4. In the second subsection of the Results, the authors report long-range interaction predictions from IntaRNA. Are these all the predictions from IntaRNA or have the authors reported the subset of the most significant predictions? It might be worthwhile to add a Supplementary table listing all the interactions that meet a reasonable significance cutoff.

5. In the paragraph above Table 1, it seems there is a typo in the second sentence. For the region 23,675-28,045, the table shows n=92 while the text says n=2.

6. There are many grammatical mistakes scattered throughout the manuscript, which should be corrected, if necessary using free software such as Grammarly.

6. PLOS authors have the option to publish the peer review history of their article (what does this mean?). If published, this will include your full peer review and any attached files.

Reviewer #1: **Yes: **Martin Raden

Reviewer #2: No

---

## [Author Response · Author response to Decision Letter 0]

26 Feb 2022

# Review for PONE-D-21-35236

Response to referees:

We are grateful to both the reviewers for their enthusiastic feedback and constructive criticism on our manuscript. We are also thankful for the helpful suggestions in improving the different sections. We have now incorporated the suggestions provided by the reviewers and hope that the revised version of the manuscript will now meet the expectations of the referees. In particular, the revised manuscript has been re-organized to address some of the major points raised by both reviewers regarding the underlying assumption of the work. In addition, we considered a bigger and a more updated dataset for our population analysis (around 200’000 rather than previously 27’000 sequences). Below are the responses to the raised queries:

Point-by-point response to reviewer’s concerns:

Reviewer #1: 

# General remarks

I have couple of central issues with this manuscript in its current form.

## (1) 12 nt insert irrelevant and obscures the focus and informative value

Beside the definition of the genomic subregion of interest, the 12 nt insert is eventually unimportant for any statement supported by the study (beside observing that S1 stem loop is stable with and without the insert).

The authors write themselves "The contribution of the 12-nt insert [...] is still unclear".

Thus, I have to ask: why then using the 12-nt fragment as such a rigid anchor for the study? The authors use about 2/3 of the Spike gene somewhat at random (based on the 12nt and its structure formation) instead of the whole gene! The rational for taking the whole could be the same, i.e. "12-nt insertion is inside", but now the SARS-CoV-1 subregion is clearly defined (as the related full gene sequence) and not based on the mapping of a substructure region...

Using just a subregion of the gene

(a) causes problems with local structure prediction AT LEAST at the ends of the regions if not overall,

(b) has thus implications on RRI prediction, since IntaRNA incorporates the accessibility of interacting subregions, which is based on local structure prediction, and

(c) restricts the insights of the study to a hypothetical local fragment rather than a valid mRNA-like molecule.

Response: We agree with the reviewer’s concern about the 12-nt-insert bias and appreciate the pertinent feedback. Hence, as suggested above, the rationale presented in the revised manuscript is now based on investigating the complete Spike region, as opposed to being exclusive to surrounding regions of the 12-nt polybasic insert. In the previous version of submission, the introduction and a great part of results (Figures 1 and 2) were based on the 12-nt polybasic insertion narrative, whereas the current version is based on the suggested rationale. Our hypothesis is now to see if SARS-CoV-2 Spike region has evolved on RNA level compared to SARS-CoV-1. The emergence of the GC-rich 12-nt polybasic insertion in SARS-CoV-2 Spike, is presented in the introduction as the background and motivation to drive a detailed analysis to understand the RNA structural evolution in Spike. The newly presented work (Figure 1 and Table 1) is the result of scanning the complete Spike region and is not biased to the 12-nt. We have also significantly revised and improved the introduction as well as most of the results sections, considering the comments from both the reviewers.

## (2) mfe structures based on global folding are no local structure signal

Using minimum free energy (mfe) estimates based on GLOBAL structure prediction (i.e. allowing base pairing over the range of the RNA) and respective structure plots to discuss local structure formation is inappropriate. The mfe is only ONE possible stable structure and strongly depends on the underlying thermodynamic model. Mfe prediction gets decreasingly reliable (in its details) the longer the RNA. Thus, instead of individual base pairing (of the presented mfe structure), LOCAL base pair probabilities (eg visualized via dot plots) are a much sounder tool to investigate local structure formation of RNAs or genomic regions. And if the details of base pairing of of no importance, even local unpaired probabilities (related to the accessibility values used by IntaRNA), are simple tool to identify locally (un)structured regions etc. without detailing base pairing.

When presenting mfe predictions of different tools (Fig-1) that are based more or less on the same thermodynamic energy model, no further insight is gained...

Furthermore, structure representations (Fig-1, Fig-4) are

-) so small that they are not readable in print

-) due to the latter, coloring is completely lost in print

-) eventually unimportant and not discussed in details for the RRI prediction

Response: We thank the reviewer for this insightful comment and agree with the reviewer’s concern that MFE-based structural specifics are not critical and can be omitted. All figures relating to RNA structure prediction are omitted from the revised manuscript as they did not carry critical information with regards to the main point of the manuscript. As suggested by the reviewer, visualizations via dot plots are now used instead, to study local structures (Figure 4). Please also note that our arguments and conclusions are based on the inferences about base-pairing probabilities rather than MFE-based base pair formation (See Discussion Paragraph 7, Locally stable RNA base pairs …). We have also improved the quality and layout the figures so that they are easily readable and interpretable. 

## (3) There is no covariation as far as I can see...

Please check Fig-1a of

https://academic.oup.com/bioinformatics/article/36/10/3072/5729989

the only thing I observe in the provided data (Fig-3) is VARIATION, i.e. only one side of the base pair shows variants...

The whole covariation part has several flaws..

(a) the authors use the concatenated interacting subsequences as an input to the CO-FOLDING BASED R-scape software... This is terribly wrong, since no LINKER sequence spacing the two FRAGMENTS is used. Without it, the helix close to the interaction that is close to the point of concatenation cannot be formed due to STERICAL reasons respected by the structure prediction software... Thus, the co-variation study misses eventually 5 base pairs and even more important, the Spike region is ONE TOO SHORT! missing the final interacting U...?!?

If done that way, ie. using the co-folding based approach, one has to insert a linker region of at least 3-5 nt, eg. poly-A or if possible poly-N, to allow for enough flexibility that the concatenated ends of the sequences can form a structure. Or why not extending the subregion range by 5nt in both directions? would be simple and avoid this central error..

Response: We agree with the reviewer about this error and have taken measures to apply the linker in the corresponding analysis. One of the suggestions was to extent the actual sequence by 5nt on both ends (totalling 20nt for each pair of segments). Original segments were extended from 23660-23-703 and 28025-28060 to 23655-23-708 and 28020-28065. Section Materials and Methods, subsection Compensatory Mutations Analysis of Long-range RNA-RNA interactions, Paragraph 2 contains the details about the modification in this revision. 

(b) who does the counts in Fig-3b relate to the substitution count in Table-1? The authors refer to the manual of R-scape, which is in no way a proper scientific explanation, even more so since these counts are central for selecting pairs and position within the argumentation of the authors..

Response: This point has now been disambiguated. Actual counts of mutations are explicitly extracted from data and presented in Figure 3 for the top four cases.

(c) the substitution counts incorporate AMBIGOUS nucleotides (Fig-3b)... ?!? What about sequencing errors etc. as a source of such ambiguity? For other parts of the study such subsequences were excluded, why not here?

Response: As suggested, we removed any ambiguity from our dataset. All remaining symbols are strictly {A,G,C,U} at all steps. Mutation counts in Figure 3 also reflect this change.

(d) the authors MISS an important point: the IntaRNA predictions are annotated with based pairs that are likely to be formed first, the seed base pairs represented by "+" within the plot. These base pairs seem to correlate with LOW substitution scores, which I would find a supportive and good sign! That is low sequence variation in the important sub-RRI!

Response: We thank the reviewer for bringing this observation to our attention. The “supportive and good sign…low sequence variation in the important sub-RRI” mentioned by the reviewer remains consistent even in our newer and bigger choice of dataset. Comparing Figures 2 and “power” column of Table 2, we can clearly see that the reviewer’s observations holds and that the more critical bonds (shown by ‘+’ in Figure 2) have lower variation (0.0 power). The above point has been mentioned in Discussion (paragraph 4, The population of SARS-CoV-2 sequences …) as one of the two main pieces of evidence for the Spike-ORF8 RNA-RNA interaction.

Thus, an investigation and relation of VARIATION with RRI formation would be more appropriate than a co-variation study that shows there is no co-variation..

Response: We agree with the reviewer on no sign of covariation in the data and have explicitly mentioned that in the discussion.

## (4) Why showing bi-/tri-molecule co-folding results?

I find no rational for nor any additional insights from the bi-/tri-molecule folding part of the manuscript. The description of the assembly of the tri-molecule prediction from bi-molecule graphs is very vague.

Maybe I missed it, but what's the point? Fig-4a shows the same RRI patterns as Fig-2a. The lack of structure relation of Fig-4b and Fig-2b can/will have multiple possible reasons that are not discussed at all..

Response: As suggested by reviewer, bi/tri-molecule co-folding results are now eliminated as they were mostly based on speculations and did not have a clear hypothesis. The corresponding figures and discussion was also omitted in the revised version.

If you are interested in co-formation of multiple RRIs, why not using a constraint IntaRNA prediction? Therein you can mark regions that are involved in base pairing (eg. an RRI) and you will get corrected predictions for the rest of the molecule EXCLUDING the ballast of local intra-molecular mfe structure (see (2)). But eventually, that wasn’t even the point, or was it? I am confused by this part..

Response: As indicated above, we have now removed the co-formation of multiple RRIs as it is not directly relevant to the core hypothesis of the manuscript and the supporting data and discussion was weak. We thank the reviewer for pointing to this weakness of this section.

## (5) No suboptimal RRIs investigated/shown/discussed ...

For RRIs, the same holds as discussed for (2). While base pairing probability prediction is much harder for RRI prediction compared to intra-molecular structure prediction, one can relate to suboptimal interactions as a substitute to assess the structural variety. This is needed at multiple parts within this manuscript:

Response: As indicated above, for each individual search, four sub-optimal predictions have been included making it a total of 5 predictions. The Supplementary Table 1, column ranking indicates the ranking of each hit. With regards to applying local stability, although the above suggestion of including local stabilities are constraint on the long-range interaction can be informative in terms of validity of the long-range binding. we did not explore this analysis in this work. Our reasoning was that the application of local stability constraints may be a very stringent constraint in initial analyses, especially for complex RNA structural mechanistic, where two regions may be in competition to bind.

(a) the extremely long UTF interaction in Fig-2b is most likely an artifact of the summation-based prediction scheme. Simply spoken: the more base pairs the better until accessibility penalties overweight the gain.

BUT: did you check suboptimal interactions of the region? I would expect much shorter sub-RRIs with similar stability (i.e. energy).

Response: We absolutely agree. Sub-optimal interactions, as suggested, were considered in the revised version. Sequence segments of 500nt (with 50nt overlap) were considered by the software, rather than applying the complete Spike gene in only one try. In addition, we used IntaRNA parameters to include top five hits rather than just one optimal hit for each run. Finally, using IntaRNA parameters, we constrained the program to exclude overlapping targets, to avoid repetitive results.

(b) Furthermore, did you check if a similar ORF8 interaction is among the sub-optimals of the SARS-CoV-1 predictions? Since the ORF8 SARS-CoV-2 RRI shows a lower energy that the UTF-RRI, the chance is high that it is "hidden" there...

Response: We thank the suggestion of the reviewer and have now investigated this possibility. However, this was not the case and even in the most inclusive sense, no hit between Spike and ORF8 in SARS-CoV-1 ever appeared. (Figure 1, and Supplementary Table 1). We find this to be a meaningful result. The observation that ORF8 appeared only in SARS-CoV-2 analysis, is detailed in the manuscript as one of the main pieces of evidence for the Spike-ORF8 RNA-RNA interaction to be specific to SARS-COV-2.

Although it may be possible to increase our prediction thresholds to generate more than 5 hits and re-investigate this further, but we doubt that this will lead to meaningful results as we already have many short-sequence hits in our results which might already be false positives. Secondly, in order to minimize the effect of length in ranking stability of interacting regions we used a generalized linear model and trained it on the results (Table 1 and Supplementary Table 1). This estimation also addresses the issue of length in the UTF-RRI (UTR-RRI) region raised by the reviewer. Indeed, the Spike-ORF8 interaction was in the top quantile of our results.

Reviewer #2: 

1. The 12-nt insertion site is at position 23,603-23,614 of the genome. The authors searched for long range interactions of any SARS-CoV-2 region with the region 23,600-24,107 and 23,917-24,118, i.e., only downstream region of the insertion site. Shouldn't equally long stretches upstream of the insertion site be considered as well? It is not clear to me why the present study has focused on the downstream region only.

Response: We completely agree with the reviewer suggestion. Hence, in the revised version, we focused on scanning the complete Spike gene which includes not only the upstream of the insert but also 50nt upstream of the start codon of Spike. We have also included 50-nt downstream of the stop codon of Spike. 

2. Similar to my previous question, it is not clear why the overlapping regions 23,600-24,107 and 23,917-24,118 were analyzed separately. It will be helpful if the authors can comment on this.

Response: In the original submission of this study, the choice of intervals was largely arbitrary. We agree with the reviewer’s comment that this forms a bias, and it is ad hoc. Hence, in the revised analysis we now equally consider all 50nt overlapping segments within Spike and have significantly revised the manuscript’s figures and results section.

3. The developers of IntaRNA have shown that integrating experimentally obtained chemical probing data with IntaRNA can significantly improve RNA-RNA interaction prediction. Since such data is available for SARS-CoV-2, have the authors considered using it in their analysis?

Response: Unfortunately, we did not consider the probing data. We agree that experimental data on full-genome of the virus (rather than on the individual sub-genomic RNAs which to our knowledge is currently available in the public domain in the form of SHAPE profiles from Anna Pyle’s lab) would have led to a more reliable long-range prediction analysis. 

4. In the second subsection of the Results, the authors report long-range interaction predictions from IntaRNA. Are these all the predictions from IntaRNA or have the authors reported the subset of the most significant predictions? It might be worthwhile to add a Supplementary table listing all the interactions that meet a reasonable significance cutoff.

Response: The supplementary Table 1 includes all possible interactions found between Spike and the rest of the genome including very weak and sub-optimal interactions. We agree that these predictions rely on assumptions and more comprehensive experimental verifications are needed to confirm their in-vivo validity as suggested by the reviewer. However, high confidence predictions as listed and discussed in our study would form a roadmap for further understanding, interpretation and validation of these theoretically sound predictions. 

5. In the paragraph above Table 1, it seems there is a typo in the second sentence. For the region 23,675-28,045, the table shows n=92 while the text says n=2.

Response: We apologize for this oversight and have corrected/updated these values.

6. There are many grammatical mistakes scattered throughout the manuscript, which should be corrected, if necessary, using free software such as Grammarly.

Response: The revised manuscript has been re-written significantly and now proofread by multiple native English-speaking colleagues.

---

## [Decision Letter · Decision Letter 1]

21 Apr 2022

PONE-D-21-35236R1Evidence for a long-range RNA-RNA interaction between ORF8 and Spike of SARS-CoV-2PLOS ONE

Dear Dr. Manzourolajdad,

Thank you for submitting your manuscript to PLOS ONE. After careful consideration, we feel that it has merit but does not fully meet PLOS ONE’s publication criteria as it currently stands. Therefore, we invite you to submit a revised version of the manuscript that addresses the points raised during the review process.

The revised manuscript should address all the critical points raised by the important reviewer.

We look forward to receiving your revised manuscript.

Kind regards,

Danny Barash

Academic Editor

PLOS ONE

Reviewers' comments:

Reviewer's Responses to Questions

**Comments to the Author**

1. If the authors have adequately addressed your comments raised in a previous round of review and you feel that this manuscript is now acceptable for publication, you may indicate that here to bypass the “Comments to the Author” section, enter your conflict of interest statement in the “Confidential to Editor” section, and submit your "Accept" recommendation.

Reviewer #1: (No Response)

2. Is the manuscript technically sound, and do the data support the conclusions?

Reviewer #1: Partly

3. Has the statistical analysis been performed appropriately and rigorously? 

Reviewer #1: I Don't Know

4. Have the authors made all data underlying the findings in their manuscript fully available?

Reviewer #1: No

5. Is the manuscript presented in an intelligible fashion and written in standard English?

Reviewer #1: Yes

6. Review Comments to the Author

Reviewer #1: # Review for PONE-D-21-35236-R1

Evidence for a long-range RNA-RNA interaction between ORF8 and Spike of SARS-CoV-2

Okiemute B. Omoru, Filipe Pereira, Sarath Chandra Janga, Amirhossein Manzourolajdad

# Summary

In short, the authors want to identify putative long-range RRIs within the SARS-CoV-2 genome that have high chance to be virus specific. To this end, they do a comparative investigation with SARS-CoV-1. Since there is a SARS-CoV-2-specific high-GC insert in Spike, the authors focus on this gene. Using a sliding window approach, putative interaction sites of Spike with other genomic regions are identified for both viruses. Since only SARS-CoV-2 shows a stable RRI with ORF8, the authors focus on the support of one of the predictions using structure prediction, base pair probabilities and (co-)variation analyses.

# General remarks

The revised work has positively sharpened the focus of the manuscript and provides more motivation for the whole endeavor. While improved, it still shows flaws and is still, in my opinion, of limited interest.

My major points are:

(*) There is no evidence anywhere that the identified RRIs are true "long-range" interactions!

The best one can speak of are "putative long-range" interactions, since (a) the whole identification and prediction is *fragment-based*, *in-silico* and without evidence that the full genome forms the interaction. It is also quite likely that the interaction is formed but only by (m)RNA fragments such that the whole hypothesis of the manuscript would be lost...

This is nowhere discussed within the manuscript!

Furthermore, both title and main contributions of the article need to be rephrased that way..

(*) The presented Pseudoknot structure prediction is neither local nor in line with the other used tools.

While it is hard to impossible to study or compare the dot-bracket-reported PK structures (Table 3), I find their presentation of no use and eventually wrong in the used context. First, the ProbKnot tool does a GLOBAL mfe prediction, with the limitations I discussed in the last review. Thus to infer local structure information from single mfes is, in my opinion, optimistic and wrong. Furthermore, all other used models within the study (IntaRNA and bp-prob computation) are based on nested structure models.. Thus, the base pair probs discussed along with the PK are using a different energy AND structure model and are thus hard to compare and a wrong intuition is triggered by the current text layout (namely that bp probs and PK structure are related).

Why using it at all? In the end, only a single minor crossing helix (5bp) is found and none for SARS-CoV-1. I doubt the relevance of this observation. Even more so while the authors have no mechanistic or whatsoever explanation or discussion of the observation (beside that it is observed).

To sanatize the course of the manuscript, I recommend to drop the PK part.

(*) Inconsistencies within the manuscript

- the Methods section still lists tons of tools that are not used in the current version

- the manuscript still referes to bifold predictions that are not present

- the 2nd and 4th paragraph of the introduction are mainly redundant

- the RRI visualizations (taken from IntaRNA) are using local fragment indices for Spike rather than the genomic positions, which makes it hard to follow and map the information. Just edit! (both in Fig-2 and supplement)

(*) Missing rationals

- the reason why the authors investigate the Spike-ORF8 RRI is only given within the discussion and one is lost wondering in the result section

- the use and interpretation of the AIC is no where to find and it stays unclear if the reported values are good or bad or how to interprete at all..

(*) Missing data

- the genome versions are not given (important since the reference genomes are undergoing some changes)

- the supplement lists the whole set of 2 million genome IDs but not the 200k used for the analyses

- how was the linear energy model derived? tool? library? handwritten?

(*) Overstating seed base pairs of RRIs

The "+" annotated bps in IntaRNA outputs are from stable subinteractions (of a used/default defined length, typically 7bp). Thus, these so called seed interactions are (in itself) stable enough to form (i.e. typically in unstructured regions) and thus likely to be starting points of the full RRI formation. Since an interaction can cover multiple such regions, all respective bps are annotated.

Currently, the manuscript describes these basepairs as "those that pair earlier than other base pairs", which is not true but again just a hypothesis.

Respective formulations should be amended respectively.

(*) Artifacts from subopt-limit

Since the authors limited the predicted suboptimals to 5 per fragment, the interaction atlas presented in Fig-1 is limited too. It could be that certain regions could interact with even more regions, which just dont pop up due to the hard "top-5" limit.

While this is no big drawback, it needs discussion. Even more so since the lack of predicted RRIs is a central point of discussion within both the result as well as discussion section!

(*) Suggestion: alpha via p-values

Just a suggestion that could improve the presentation. Given a large amount of genome-wide predictions (as done here) it is possible to estimate p-values for the energy scores (as presented on the IntaRNA webserver). The IntaRNA package even provides a respective script for computation.

The p-values could be used to set the alpha channel of the arcs within the circle plots to highlight highly probable RRIs and to distinguish them from weaker ones.

Currently, it is hard to say what interactions are stable. Even more so, since there is no general energy cut-off etc.

(*) Fig 4 not interpretable in printed version (and hard in pdf)

The dot plots are so small that dots are hard to spot or interprete/compare in print.

But even in pdf this is hard since only a pixel graphics is provided that cannot be zoomed without getting pixelated.

I suggest to move both figures in vector graphics format (PDF) to the supplement in full page width each to allow for detailed investigation.

The authors could present a respective cutout for the main manuscript if needed.

(*) Minor issues that caught the eye

- the text uses "Orf.." instead of "ORF.."

- Table 1 (and the supplement table) do not use "ORF8" but rather just "8" etc.

- Table 1 shows a red highlight not discussed in its caption

- "11th top hit within a total of 66" .. what 66? or is about the 69?

- "segments each corresponding to a particular viral strain" .. nope, each corresponds to a full genome sequencing (i.e. sample) but not necessarily strain!

- Table 2: lines Spike 23679-80 should be bold too

- it is not discussed that the CaCoFold predictions (Table 2 + Fig 3) miss the left-most RRI part from Fig-2, thus, rendering that RRI part less likely

- Fig-3 seems to be of low quality

- no vector graphics.. zoom in provides pixel art ..

- "contains five pseudoknots" .. NO, only ONE KNOT but "5 crossing base pairs". A BIG difference!

- Fig-4: it would be helpful to state in the caption that most likely base pairs are colored in red (for non-math readers)

- the text often states "SARS-CoV" instead of "SARS-CoV-1"

- Fig-4 caption "using the partition function" is a useless comment. better name the used tool.

- Fig-4 "shown in bold" .. better use "highlighted with a black bar". You can also annotate the same region on the y-axis (same coordinates) and draw horizontal/vertical lines at the respective bar ends to guide the eye to the important corridors within the plot

- "as well as well"

- it would be interesting to relate the S1 hairpin with the dot plot or annotate it(s position) within

- the supplementary figure needs a caption or the figure has to be extended to be self-explaining (what is relating to what and where)

(*) Carving out the core RRI based on the variation and stability investigation

Eventually, I think the authors miss a central outcome of the study or do not present it as such. While all the "long-range" part and the hopeful hypothesis of its impact on virulence, regulation etc. is quite speculative, the authors miss to highlight that the integration of RRI prediction and variation analyses strongly identifies the core of the putate Spike-ORF8 interaction. Namely the seed-region stretch 23679-23690 (Spike). The left part of Fig-2 is not predicted in Fig-3 and no variation is seen in this area.

Thus, one can conclude that the true RRI part (or at least the most likely part) is defined by that region and that it might be relevant (but maybe not exclusively for the RRI) since it is not mutated in both genes.

Why is it that most core conclusions of that manuscript are by reviewers?

7. PLOS authors have the option to publish the peer review history of their article (what does this mean?). If published, this will include your full peer review and any attached files.

Reviewer #1: **Yes: **Martin Raden

---

## [Author Response · Author response to Decision Letter 1]

7 Jun 2022

Response to Reviewer’s Comments

Reviewer #1: # Review for PONE-D-21-35236-R1

Evidence for a long-range RNA-RNA interaction between ORF8 and Spike of SARS-CoV-2

Okiemute B. Omoru, Filipe Pereira, Sarath Chandra Janga, Amirhossein Manzourolajdad

# Summary

In short, the authors want to identify putative long-range RRIs within the SARS-CoV-2 genome that have high chance to be virus specific. To this end, they do a comparative investigation with SARS-CoV-1. Since there is a SARS-CoV-2-specific high-GC insert in Spike, the authors focus on this gene. Using a sliding window approach, putative interaction sites of Spike with other genomic regions are identified for both viruses. Since only SARS-CoV-2 shows a stable RRI with ORF8, the authors focus on the support of one of the predictions using structure prediction, base pair probabilities and (co-)variation analyses.

# General remarks

The revised work has positively sharpened the focus of the manuscript and provides more motivation for the whole endeavor. While improved, it still shows flaws and is still, in my opinion, of limited interest.

My major points are:

(*) There is no evidence anywhere that the identified RRIs are true "long-range" interactions!

The best one can speak of are "putative long-range" interactions, since (a) the whole identification and prediction is *fragment-based*, *in-silico* and without evidence that the full genome forms the interaction. It is also quite likely that the interaction is formed but only by (m)RNA fragments such that the whole hypothesis of the manuscript would be lost...

This is nowhere discussed within the manuscript!

Furthermore, both title and main contributions of the article need to be rephrased that way..

Response: We thank the reviewer for these suggestions and have now revised the title and discussion to reflect this input. The main hypothesis is the predicted Spike-ORF8 interaction. Various mechanistic speculations as discussed by the reviewer are mentioned in the discussion. The in-silico fragment-based nature of our approach is also emphasized in Abstract, Introduction, and Discussion. The main contribution of the work is further clarified. We thank the reviewer for encouraging us to clarify our main finding regarding integration of thermodynamic-based modeling and mutation patterns to identify the core sub-interacting region in the Spike-ORF8 prediction.

(*) The presented Pseudoknot structure prediction is neither local nor in line with the other used tools.

While it is hard to impossible to study or compare the dot-bracket-reported PK structures (Table 3), I find their presentation of no use and eventually wrong in the used context. First, the ProbKnot tool does a GLOBAL mfe prediction, with the limitations I discussed in the last review. Thus to infer local structure information from single mfes is, in my opinion, optimistic and wrong. Furthermore, all other used models within the study (IntaRNA and bp-prob computation) are based on nested structure models.. Thus, the base pair probs discussed along with the PK are using a different energy AND structure model and are thus hard to compare and a wrong intuition is triggered by the current text layout (namely that bp probs and PK structure are related).

Why using it at all? In the end, only a single minor crossing helix (5bp) is found and none for SARS-CoV-1. I doubt the relevance of this observation. Even more so while the authors have no mechanistic or whatsoever explanation or discussion of the observation (beside that it is observed).

To sanatize the course of the manuscript, I recommend to drop the PK part.

Response: As recommended by the reviewer, we have now removed the PK part from the manuscript to improve the clarity and flow of the manuscript. 

(*) Inconsistencies within the manuscript

- the Methods section still lists tons of tools that are not used in the current version

Response: We have now removed the listing of tools in the methods section which are not used in the current version of the manuscript. 

- the manuscript still refers to bifold predictions that are not present

Response: We have now removed the bifold predictions from the manuscript.

- the 2nd and 4th paragraph of the introduction are mainly redundant

Response: Thank you for this suggestion. We have now reduced the redundancy in these paragraphs of the introduction. 

- the RRI visualizations (taken from IntaRNA) are using local fragment indices for Spike rather than the genomic positions, which makes it hard to follow and map the information. Just edit! (both in Fig-2 and supplement)

Response: We appreciate the reviewer pointing to this inconsistency and have now edited the figure 2 to reflect the genomic positions so that it is easy to map and follow the information across the study. 

(*) Missing rationals

- the reason why the authors investigate the Spike-ORF8 RRI is only given within the discussion, and one is lost wondering in the result section

Response: To improve the flow and logic for investigating Spike-ORF8 RRI in this study, we have now included a transition in the result section right before the Spike-ORF8 section providing a rational for choosing these regions for performing RRI analysis in this study.

- the use and interpretation of the AIC is no where to find and it stays unclear if the reported values are good or bad or how to interprete at all..

Response: We have now removed AIC from the main text. As a measure of model fitness, we used the significant factor of the Length parameter instead. (Pr(>|t|) for length is reported in Table 1 caption along with other details of the model used. We have also explained details regarding model derivation.

(*) Missing data

- the genome versions are not given (important since the reference genomes are undergoing some changes)

Response: We have now included the specific version of the genomes that were used in the study. 

- the supplement lists the whole set of 2 million genome IDs but not the 200k used for the analyses

Response: As suggested by the reviewer, we have now included the genome sequence IDs for the 200K genomes that were used in the analyses. 

- how was the linear energy model derived? tool? library? handwritten?

Response: The linear energy model was generated in R statistical package, and we now have explained it in the manuscript. 

(*) Overstating seed base pairs of RRIs

The "+" annotated bps in IntaRNA outputs are from stable subinteractions (of a used/default defined length, typically 7bp). Thus, these so called seed interactions are (in itself) stable enough to form (i.e. typically in unstructured regions) and thus likely to be starting points of the full RRI formation. Since an interaction can cover multiple such regions, all respective bps are annotated.

Currently, the manuscript describes these basepairs as "those that pair earlier than other base pairs", which is not true but again just a hypothesis.

Respective formulations should be amended respectively.

Response: We have now changed multiple places in the manuscript to reflect the above interpretation recommended by the reviewer. We refrained from speculations regarding the above results. 

(*) Artifacts from subopt-limit

Since the authors limited the predicted suboptimals to 5 per fragment, the interaction atlas presented in Fig-1 is limited too. It could be that certain regions could interact with even more regions, which just dont pop up due to the hard "top-5" limit.

While this is no big drawback, it needs discussion. Even more so since the lack of predicted RRIs is a central point of discussion within both the result as well as discussion section!

Response: As suggested by the reviewer, we have now elaborated the discussion to reflect on the possibility that there could be additional RRIs which may have escaped our limit of top 5 hits but still could be biologically interesting. 

(*) Suggestion: alpha via p-values

Just a suggestion that could improve the presentation. Given a large amount of genome-wide predictions (as done here) it is possible to estimate p-values for the energy scores (as presented on the IntaRNA webserver). The IntaRNA package even provides a respective script for computation.

The p-values could be used to set the alpha channel of the arcs within the circle plots to highlight highly probable RRIs and to distinguish them from weaker ones.

Currently, it is hard to say what interactions are stable. Even more so, since there is no general energy cut-off etc.

Response: We agree that p-values and/or setting cut-off are also good methods for producing meaningful results. In this work, however, we had decided to use an alternative approach for ranking predictions, which is the residual values of our model as discussed in results section.

(*) Fig 4 not interpretable in printed version (and hard in pdf)

The dot plots are so small that dots are hard to spot or interprete/compare in print.

But even in pdf this is hard since only a pixel graphics is provided that cannot be zoomed without getting pixelated.

I suggest to move both figures in vector graphics format (PDF) to the supplement in full page width each to allow for detailed investigation.

The authors could present a respective cutout for the main manuscript if needed.

Response: We have now made a concerted effort to significantly increase the resolution of Figure 4 for better readability. However, please note that PLoS One submission system often decreases the resolution of submitted figures for peer review purposes to generate less heavy files for reviewers and it may be possible that this has resulted in down resolution. Nevertheless, as suggested we have now also included the figure as supplementary file too. 

(*) Minor issues that caught the eye

- the text uses "Orf.." instead of "ORF.."

- Table 1 (and the supplement table) do not use "ORF8" but rather just "8" etc.

- Table 1 shows a red highlight not discussed in its caption 

- "11th top hit within a total of 66" .. what 66? or is about the 69? 

- "segments each corresponding to a particular viral strain" .. nope, each corresponds to a full genome sequencing (i.e. sample) but not necessarily strain! 

- Table 2: lines Spike 23679-80 should be bold too 

- it is not discussed that the CaCoFold predictions (Table 2 + Fig 3) miss the left-most RRI part from Fig-2, thus, rendering that RRI part less likely

- Fig-3 seems to be of low quality

- no vector graphics.. zoom in provides pixel art ..

- "contains five pseudoknots" .. NO, only ONE KNOT but "5 crossing base pairs". A BIG difference!

- Fig-4: it would be helpful to state in the caption that most likely base pairs are colored in red (for non-math readers) 

- the text often states "SARS-CoV" instead of "SARS-CoV-1"

- Fig-4 caption "using the partition function" is a useless comment. better name the used tool. 

- Fig-4 "shown in bold" .. better use "highlighted with a black bar". You can also annotate the same region on the y-axis (same coordinates) and draw horizontal/vertical lines at the respective bar ends to guide the eye to the important corridors within the plot

- "as well as well"

- it would be interesting to relate the S1 hairpin with the dot plot or annotate it(s position) within

- the supplementary figure needs a caption or the figure has to be extended to be self-explaining (what is relating to what and where) 

Response: We sincerely thank the reviewer for these suggestions and have now made every effort to address all these minor issues to significantly clean up the manuscript.

(*) Carving out the core RRI based on the variation and stability investigation

Eventually, I think the authors miss a central outcome of the study or do not present it as such. While all the "long-range" part and the hopeful hypothesis of its impact on virulence, regulation etc. is quite speculative, the authors miss to highlight that the integration of RRI prediction and variation analyses strongly identifies the core of the putate Spike-ORF8 interaction. Namely the seed-region stretch 23679-23690 (Spike). The left part of Fig-2 is not predicted in Fig-3 and no variation is seen in this area.

Thus, one can conclude that the true RRI part (or at least the most likely part) is defined by that region and that it might be relevant (but maybe not exclusively for the RRI) since it is not mutated in both genes.

Why is it that most core conclusions of that manuscript are by reviewers?

Response: We appreciate the input from the reviewer, but we want to emphasize that study’s main goal was to identify the core RRIs in the Spike-ORF8 regions as the main contribution. Co-variation analysis was initially anticipated to be an independent means for understanding the functional meaning and evolutionary conservation of these inferred associations.

---

## [Decision Letter · Decision Letter 2]

23 Jun 2022

A Putative long-range RNA-RNA interaction between ORF8 and Spike of SARS-CoV-2

PONE-D-21-35236R2

Dear Dr. Manzourolajdad,

We’re pleased to inform you that your manuscript has been judged scientifically suitable for publication and will be formally accepted for publication once it meets all outstanding technical requirements.

Kind regards,

Danny Barash

Academic Editor

PLOS ONE

Additional Editor Comments (optional):

Reviewers' comments:

Reviewer's Responses to Questions

**Comments to the Author**

1. If the authors have adequately addressed your comments raised in a previous round of review and you feel that this manuscript is now acceptable for publication, you may indicate that here to bypass the “Comments to the Author” section, enter your conflict of interest statement in the “Confidential to Editor” section, and submit your "Accept" recommendation.

Reviewer #1: All comments have been addressed

2. Is the manuscript technically sound, and do the data support the conclusions?

Reviewer #1: Yes

3. Has the statistical analysis been performed appropriately and rigorously? 

Reviewer #1: Yes

4. Have the authors made all data underlying the findings in their manuscript fully available?

Reviewer #1: Yes

5. Is the manuscript presented in an intelligible fashion and written in standard English?

Reviewer #1: Yes

6. Review Comments to the Author

Reviewer #1: (No Response)

7. PLOS authors have the option to publish the peer review history of their article (what does this mean?). If published, this will include your full peer review and any attached files.

Reviewer #1: **Yes: **Martin Raden

---

## [Editor Report · Acceptance letter]

20 Jul 2022

PONE-D-21-35236R2 

A Putative long-range RNA-RNA interaction between *ORF8* and *Spike* of SARS-CoV-2 

Dear Dr. Manzourolajdad:

I'm pleased to inform you that your manuscript has been deemed suitable for publication in PLOS ONE. Congratulations! Your manuscript is now with our production department. 

Kind regards, 

on behalf of

Dr. Danny Barash 

Academic Editor

PLOS ONE